# Experiences of health providers regarding implementation of the physiologic birth program in Iran: A qualitative content analysis

**Azam Moridi[1], Parvin Abedi[2]\*, Mina Iravani[2], Shala Khosravi[3], Narges Alianmoghaddam[4], Elham Maraghi[5], Najmieh Saadati[6]**

1 Department of Midwifery, Nursing and Midwifery School, Ahvaz Jundishapur University of Medical Sciences, Ahvaz, Iran, 2 Department of Midwifery, Reproductive Health Promotion Research Center, Ahvaz Jundishapur University of Medical Sciences, Ahvaz, Iran, 3 Department of Community Medicine, Faculty Member of Medicine School, Tehran University of Medical Sciences, Tehran, Iran, 4 School of Public Health, Massey University, Palmerston North, New Zealand, 5 Department of Biostatistics and Epidemiology, Faculty of Public Health, Ahvaz Jundishapur University of Medical Sciences, Ahvaz, Iran, 6 Obstetrics and Gynecology, Fertility Infertility and Perinatology Research Center, Ahvaz Jundishapur University of Medical Sciences, Ahvaz, Iran

\* parvinabedi@ymail.com

## Abstract

### Introduction

The rate of cesarean section is on the rise in both developed and developing countries, and Iran is no exception. According to the WHO, physiologic labor is one of the main strategies for reducing cesarean section and improving the health of mothers and newborns. The aim of this qualitative study was to explain the experiences of health providers regarding implementation of the physiologic birth program in Iran.

### Methods

This study is a part of a mixed-methods study, in which 22 health providers were interviewed from January 2022 to June 2022. Data analysis was performed using Graneheim and Lundman's conventional content analysis approach and using MAXQDA10 software.

### Results

Two main categories and nine subcategories emerged from the results of this study. The main categories included "the obstacles to the implementation of the physiologic birth program" and "strategies for improving implementation of the program". The subcategories of the first category included: lack of continuous midwifery care in the healthcare system, lack of free accompanying midwives, lack of integrated healthcare and hospitals in service provision, low quality of childbirth preparation and implementation of physiologic birth classes, and lack of requirements for the implementation of physiologic birth in the maternity ward. The second category included the following subcategories: Supervising the implementation of childbirth preparation classes and physiologic childbirth, support of midwives by

**Data Availability Statement:** Data availability statement: In order to preserve participants' confidentiality, and according to the requirements

of the METC Groningen in which anonymity of participants must be guaranteed, we are not willing to share the qualitative datasets (the interview transcripts) in the main paper or additional supporting files. We cannot share the data due to ethical restrictions that the data contains potentially identifiable and sensitive information of the participants. Although we did remove personal identifiers from the interview transcripts (e.g. names and addresses), the transcripts are likely to contain references to the contextual identifiers in individual stories and make individuals identifiable. When providing their informed consent to participate in the study, participants were ensured their privacy would be protected. They did not provide consent for their data to be shared in a repository. We can provide access to the transcripts and audit trail on request and subject to certain conditions. Data requests must be addressed to the Reproductive Health Promotion Research Center of Ahvaz Jundishapur University of Medical Sciences, that will provide access after evaluating requests: RHPRC@ajums.ac.ir.

**Funding:** The expenses of this study were provided by Ahvaz Jundishapur University of Medical Sciences. The funders had no role in study design, data collection and analysis, decision to publish, or preparation of the manuscript.

**Competing interests:** The authors have declared that no competing interests exist.

insurance companies, holding training courses on physiologic birth, and evaluation of program implementation.

## Conclusions

The experiences of the health providers with the physiologic birth program revealed that policymakers should provide the ground for the implementation of this type of labor by removing the obstacles and providing the particular operational strategies needed in Iran. Important measures that can contribute to the implementation of the physiologic labor program in Iran include the following: Setting the stage for physiologic birth in the healthcare system, creating low- and high-risk wards in maternity hospitals, providing professional autonomy for midwifery, training childbirth providers on physiologic birth, monitoring the quality of program implementation, and providing insurance support for midwifery services.

## Introduction

According to the latest report of the World Health Organization (WHO), 21.1% of women worldwide gave birth by caesarean, and the rate of cesarean section is 45.5% in Iran [1]. In private hospitals, this rate is greater than 60% [2]. According to the statement of the WHO, the ideal cesarean rate should be between 10 and 15 percent. According to Iran's fifth development plan (2011), the rate of cesarean section was supposed to be decreased by 10% by the end of 2014, but almost all Iranian hospitals did not reach this rate [3]. As shown in different studies, a 10% reduction in the rate of caesarean section in a society means a reduction in maternal and newborn mortality during childbirth. The increased rate of mortality during caesarean section is 4 to 5 times higher than that of normal vaginal delivery [4]. Mortality rate is 2.1 per 100,000 cases in vaginal delivery, 5.9 in elective cesarean, and 18.2 in emergency cesarean [5].

The physiologic birth program was implemented in line with the program for the promotion of vaginal delivery and with the aim of improving maternal and newborn health and reducing the rate of cesarean session. During the recent decade, the Iranian Ministry of Health followed these objectives by holding childbirth preparation classes for pregnant mothers and empowering the health providers [6]. Based on the guidelines of the WHO, this program is aimed to lower the unnecessary medical interventions in the natural process of childbirth, provide personal support for women by a reliable person, increase the freedom of movement during labor, encourage the use of positions other than supine and increase the mother-infant skin-to-skin contact immediately after delivery and during breastfeeding [7]. A physiologic labor and birth is one that is powered by the innate human capacity of the woman. This type of birth is more likely to be safe and healthy because there is no unnecessary intervention to disrupt physiologic processes [8]. Furthermore, in physiologic labor, a specific emphasis is put on reduction of labor pain using non-pharmacological methods such as massage therapy, aromatherapy, heat and cold therapy, acupressure, music therapy, reflexology, relaxation, and breathing techniques [9, 10].

The Lamaze Method involves six care techniques for the support and promotion of physiologic labor and recommends them to all childbirth care providers [11]. These care techniques include spontaneous initiation of labor, freedom of movement in labor, being supported by a companion, avoiding usual medical interventions, spontaneous pushing in vertical positions and using the same room for the mother and her newborn, and facilitating the position of

breastfeeding [12]. Physiologic labor program is designed based on the human rights of mothers. These services can be provided by maternity midwives or private midwives in hospitals, although midwives working in hospitals are not allowed to serve as simultaneous private midwives and duty personnel in the same shift. In this type of labor, attention is paid to the empowerment of mothers and their freedom of action and self-control. Women's ability to feel self-control during the labor process is the basis of a positive childbirth experience [13].

According to the announcement of the Iranian Ministry of Health, Iran had the second highest rank in the rate of caesarean section (54%) worldwide in 2014 [14]. With the implementation of the physiologic labor program since 1993, the results of studies have shown that the cesarean rate in Iran has decreased by 6% immediately after the implementation of this program and then leveled off [15]. Moreover, the rate of vaginal delivery rate remained unchanged (57%). Based on Iran's development program, the physiologic birth program could not effectively reduce the cesarean rate in this country according to predetermined goals, even in public hospitals [16]. This is indicative of the existence of obstacles and challenges in the implementation of this program. Therefore, it is imperative to investigate the success of the physiologic birth program in the healthcare system [17]. As such, this study was conducted to investigate the experiences of health care providers about the physiologic birth program in Iran.

The impact of the COVID-19 pandemic on the health system and the subsequent birth preparation classes cannot be ignored. Studies have shown a reduction in the number of prenatal care visits followed by a reduction in attendance to health centers during the COVID-19 pandemic [18]. A study in Iran showed that the fear of transmitting the disease caused many mothers not to go to health centers for prenatal care. Worth mentioning that childbirth preparation classes were held continuously through online classes during COVID-19 pandemic in Iran [19].

## Methods

### Study design and setting

Using conventional a qualitative content analysis approach, this study was conducted to achieve a deep understanding of the participants' experiences. To this aim, the physiologic birth experiences of health providers in the health care system in Ahvaz Jundishapur University of Medical Sciences (health centers that organize childbirth preparation classes and Sina Hospital, where childbirth is attended only by midwives) were assessed from January to June 2022. This study is a part of a mixed-methods study entitled "Providing a physiological delivery intervention program in Iran's health system: An embedded mixed-methods study. While doing content analysis, the researcher allows subcategories and categories to emerge from the data, and hence, they adhere to the naturalistic paradigm [20]. This paradigm leads to the emergence of deep and profound data that can clarify various dimensions of complex human phenomena [21]. The physiologic birth program has been implemented for more than a decade in Iran but has been accompanied with certain obstacles. In order to remove these obstacles, it is essential to explore the experiences of influential people within this program and to adopt effective strategies for improving its quality.

### Ethical considerations

This study was approved by the Ethics Committee of Ahvaz Jundishapur University of Medical Sciences (Ref. ID: **IR.AJUMS.REC.1401.050**). Confidentiality of the collected data was ensured, and written informed consent was obtained from all research participants. Also, participants were free to decline participation or withdraw at any stage of the research process. All interviews were recorded with the permission of the participants, and all the audio files were securely stored in password-protected computers.

## Data collection and participants

The participants in this study included 23 health providers who met the inclusion criteria of having a certificate of attending 60 hours of physiologic birth educational classes and having at least 5 years of clinical work experience. In order to select the participants, purposive sampling method was used and continued until data saturation; that is, until no new information or data was revealed. Purposive sampling was done considering maximum variety in terms of the participants' workplace, work experience (years), educational attainment, age, etc. For data collection, in-depth and semi-structured individual face-to-face interviews were conducted after obtaining informed consent from the participants. Due to the maximum variety of the participants who were from different levels of service providers, including managerial, executive, clinical, and educational levels in both public and private sectors, the interviews were expected to provide a deep understanding of the phenomenon of physiological childbirth from the participants' point of view. All interviews were conducted by first investigator that is PhD student in Midwifery (AM). This researcher passed necessary courses on qualitative studies and also conducted some interviews under the supervision of the second and third authors. Before interview, the lead researcher introduced herself to participants, presented the objectives of the study, and why she is interested in this topic. Before the outset of the interview, the interviewer attempted to communicate with the participants and create a friendly atmosphere by introducing herself and talking to the participants and answering their questions. Then, at the beginning of the interview and prior to its formal commencement, the researcher gave a short verbal explanation about the reasons and objectives of the study and answered any possible question of the participants. This led to the establishment of a good relationship and trust between them. Finally, the participants were assured about their freedom in answering the questions and participating in the research, confidentiality of information, and the possibility of withdrawal from the study at any stage. Given the lead researcher's interest and work experience in the field of physiologic birth, she conducted the interview, data collection, and data analysis by leaving aside her previous thoughts and assumptions.

The place and time of conducting the interview were determined at the participants' convenience. The interviews were conducted without time limit in a separate room at the health centers or the hospital where only the interviewer and the participant were present. The interviews continued until all relevant experiences of the participants were fully explained and comprehensive answers were received from the interviewees. The interviews were recorded using a digital audio recorder. The participants' consent for recording the interview was obtained, and in case they did not allow the recording, field notes were taken.

In semi-structured interviews, the questions are not fixed and predetermined, but are formed based on the interview process. To start the interview based on the aim of the study, the following general and open question was first asked: "Could you please tell us about your experience with the physiologic birth program?" Then the following questions were posed: "What obstacles do you see in the implementation of the program?" and "What are your solutions for the better implementation of the program?" During the interview, in-depth and probing questions were asked based on the type of answer to each question in order to explore the depth of the experience: "What do you mean?", "Why?", Please elaborate on that", and "Could you give an example so that I understand what you mean." During the interview, the researcher recorded non-verbal data such as the moods of the participants and their paralinguistic features such as tone of voice, facial expressions, and posture. Each interview lasted from 45 to 60 minutes.

The interviews continued until data saturation. Data saturation was achieved after the 20th interview, but 3 more interviews were conducted to ensure saturation. Finally, 23 individual

in-depth interviews were conducted with health providers. No participant was excluded/ dropped out from the study, nor was any interview repeated.

## Data analysis

Conventional content analysis was used for data analysis. The data analysis process was performed according to the steps suggested by Graneheim and Lundman [22]. Immediately after conducting each interview, the recorded interview was transcribed. Then the entire text was read for a general understanding of its content, determining the meaning units and primary codes, and classifying similar primary codes into more comprehensive categories. At the earliest possible time after conducting the interview (generally a few hours after the end of the interview), the lead researcher transcribed the interview verbatim and performed data analysis. Then, the entire text was read several times to get a general understanding of the content of the interview. The entire text was considered as the unit of analysis, and smaller parts including the words, phrases, sentences, or paragraphs which had a meaning or concept related to the research question, were considered as a meaning unit. Each meaning unit was first converted into condensed meaning units by keeping the original concept, and then, these units were coded. The coding process was performed by two authors (AM, PA). The codes were classified into subcategories and categories based on their similarities and differences. The data analysis process was performed using MAXQDA software (version 10).

In the present study, the four criteria of Lincoln and Guba [23] were followed to increase the trustworthiness of the data. The credibility of the research data was ensured by the continuous involvement of the researcher with the subject of the research and spending enough time to collect the data. Moreover, the content of the categories was reviewed by two participants and the researchers to ensure the correspondence between the categories obtained and the experiences of the participants. The participant reviewed a short report of the analyzed data for member check to see how well the report reflected their experiences and attitudes.

Dependability was ensured using the opinions of external observers (two experts in midwifery and reproductive health) as well as code-recode method during the analysis. Transferability of the findings was obtained through a detailed description of the context, participants, environment, and conditions. To ensure conformability, the lead researcher who did the analysis excluded her presuppositions and thoughts and used the opinions of two midwifery and reproductive health experts to reach a consensus on the process of forming the subcategories and categories.

## Results

The findings were extracted from the analysis of 23 in-depth individual interviews with health providers including four childbirth preparation class instructors, four maternity midwives, four doulas, three gynecologists, four national senior lecturers of physiologic labor, the head of the family health department, maternal health officer, a midwife in charge of the maternity hospital, and a supervisor. The characteristics of the participants are presented in Table 1.

### 1. Obstacles to implementation of the physiologic birth program

1. This main category included 5 subcategories as follows:

**1.1. No continuous midwifery care in the healthcare system.** The experience of the participants was indicative of the dire need for continuous midwifery care in Iran's health system. *The best situation is that the midwife, who has been with the mother in classes and during the education and whom the mother trusts, be by her side during the labor, and when the mother has pain, she supports her until after delivery (P10, 34y, Childbirth preparation class instructor). It's*

**Table 1. Characteristics of maternity care providers.**

| Participant | Age (y) | Education | Workplace | Work experience (years) |
|---|---|---|---|---|
| 1 | 53 | Bachelor | Supervisor | 30 |
| 2 | 45 | Bachelor | Midwife in charge of the maternity hospital | 14 |
| 3 | 38 | Master | Staff midwife | 10 |
| 4 | 39 | Master | Staff midwife | 13 |
| 5 | 57 | Bachelor | Staff midwife | 29 |
| 6 | 49 | Bachelor | Doula | 24 |
| 7 | 39 | Bachelor | Staff midwife | 15 |
| 8 | 55 | Master | Doula | 28 |
| 9 | 36 | Bachelor | Doula | 7 |
| 10 | 34 | Bachelor | Childbirth preparation class instructor | 27 |
| 11 | 43 | Bachelor | Childbirth preparation class instructor | 12 |
| 12 | 54 | Bachelor | Maternal health officer | 20 |
| 13 | 57 | General Practitioner | Head of the family health department | 22 |
| 14 | 61 | PhD in Reproductive Health | National senior lecturer of physiologic childbirth | 30 |
| 15 | 49 | Gynecologist | Assistant Professor in Obstetrics & Gynecology | 25 |
| 16 | 45 | Bachelor | Childbirth preparation class instructor | 21 |
| 17 | 59 | PhD in Reproductive Health | National senior lecturer of physiologic childbirth | 30 |
| 18 | 57 | Gynecologist | Assistant Professor in Obstetrics & Gynecology | 24 |
| 19 | 59 | PhD in Reproductive Health | National senior lecturer of physiologic childbirth | 28 |
| 20 | 40 | PhD in Reproductive Health | National senior lecturer of physiologic childbirth | 17 |
| 21 | 52 | Gynecologist | Assistant professor in Obstetrics & Gynecology | 23 |
| 22 | 41 | Bachelor | Doula | 18 |
| 23 | 40 | Bachelor | Childbirth preparation class instructor | 16 |

In the analysis of interviews, 121 primary codes were extracted. After merging similar codes, 91 primary codes were obtained. After their integration, the codes were finally classified into 2 main categories and 9 subcategories. The main categories were "obstacles to the implementation of the physiologic birth program" and "strategies for improving the implementation of the program in Iran" (Fig 1).

*happened many times when the mother asks me to be beside her during the delivery and I say no. I'm just a trainee and I send her to private offices. Now mothers like to go to a midwife who will be with them from pregnancy to delivery (P11, 43y, Childbirth preparation class instructor).*

**1.2. Lack of accompanying midwives (doulas).** According to the participants, the absence of accompanying midwives in the public health system was another obstacle to implementation of the physiologic birth program. *There should be accompanying midwives in the public health system. Midwives working in health centers should receive the same overtime pay as do their counterparts working in hospitals. My shifts during the COVID-19 pandemic were evening shifts. I received extra payment for the pandemic condition. They can assign this payment for physiologic birth too (P16, 45y, Childbirth preparation class instructor).*

*My educational classes may have influenced the choice of vaginal delivery by 5%. She's received some education here; then, she goes to the maternity hospital and none of this can be useful there, and she doesn't have money to have a private midwife; it means nothing; something needs to be done (P23, 40y, Childbirth preparation class instructor).*

**1.3. Lack of integrated healthcare and treatment in service provision.** According to the participants, lack of integration between childbirth preparation classes and the continuation of the mother's labor process was one of the major obstacles to physiologic birth. *The fact that*

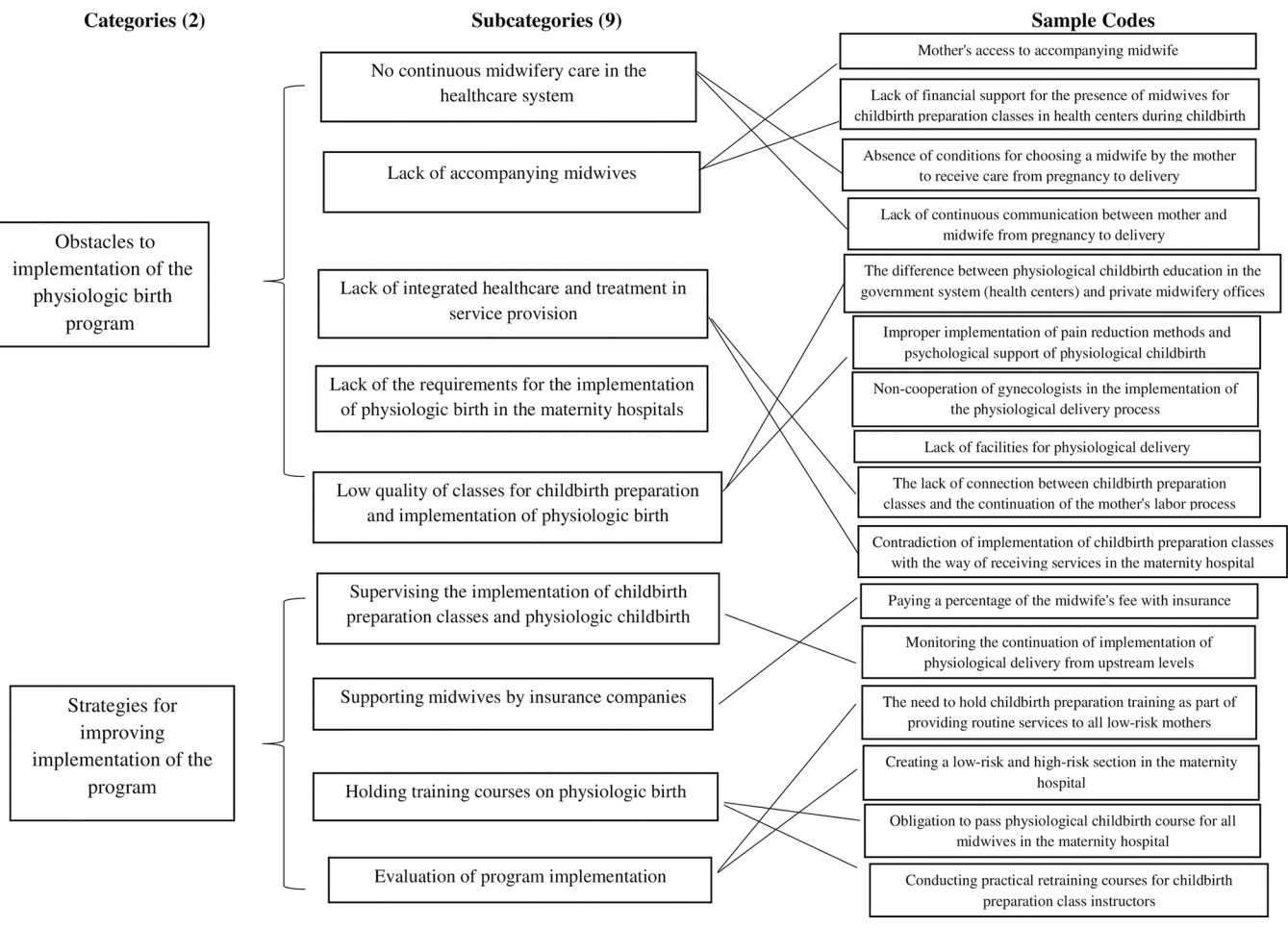

**Fig 1. The process of categorizing codes.**

*there's no integration between the childbirth preparation classes and the mother's labor is another obstacle in itself. It means that this program isn't systematic and continuous. She receives the education, but then there will be another program for her delivery (P12, 54y, Maternal health officer). The health system holds some classes which of course aren't welcomed recently; there's no planning for the mother's delivery. After classes, the mother goes to the maternity hospital, and the things taught to her aren't implemented. If we want to implement this program, the healthcare system and hospitals should decide together on how to implement it (P3, 38y, Staff midwife).*

**1.4. Low quality of classes for childbirth preparation and implementation of physiologic birth.** As far as the childbirth preparation classes were concerned, the participants were of the opinion that the following were the main obstacles to implementation of the physiologic program: No updating of the resources of childbirth preparation classes, different physiologic birth education in the public health centers vs. private midwifery offices, and incorrect implementation of pain reduction methods and psychological support by private accompanying midwives. *Information and books should be updated. This book is 2016. Of course, we don't limit ourselves to this book; we study other sources too, but these sources need to be updated (P11, 43y, Childbirth preparation class instructor).*

*I've a client who has attended the classes held by the health center. After attending my classes, she says they're completely different. What is important is the quality of the classes, not*

*providing the mother with a series of points that won't help her in childbirth. For instance, I teach her how to check the contractions when they begin. Her husband learns to put his hand on her stomach and check the contraction and informs me about it. Do they in the health center teach the mother in this way and with such sensitivity? (P8, 55y, Doula).*

*Physiologic birth is not implemented properly even by a private midwife or a companion who comes to the hospital. There is only a birth ball with low quality. They give the mother saffron syrup with dates and think that this is what physiologic birth is all about, while it's not physiologic labor at all. I used to go beside their beds and tell them to take a specific position or do a specific thing. Some colleagues still do not have enough knowledge (P4, 39y, Staff midwife). One of the important reasons for having a companion midwife is to manage pain, create distraction and provide emotional support for the mother; but unfortunately, it is not implemented at all. There is no physiologic birth at all as we expect (P2, 45y, Midwife in charge of the maternity ward).*

**1.5. Lack of the requirements for the implementation of physiologic birth in the maternity hospital.** Crowded maternity hospitals and the prioritized care of high-risk mothers were other obstacles to the implementation of physiologic birth, according to the midwives of the maternity hospital. *Midwives have a different condition in the maternity hospitals because a midwife isn't in charge of just one mother; she may have 4 or 5 patients simultaneously; she certainly can't provide a complete physiologic labor for a mother. Priority is given to high-risk mothers who need more care (P5, 57y, Staff midwife).*

Another midwife talked about the lack of facilities: *There aren't sufficient facilities; for instance, there's only one shower and five pregnant women; which one should be sent? They all need to take a hot shower. They all need to sit on the birth ball, and you've only one birth ball (P7, 39y, Staff midwife).*

According to the participants, obstetricians regarded childbirth as an unnatural process, and this was a serious obstacle to the implementation of the physiologic birth program. *The largest part of my problem is because of the obstetricians' intervention. We'd like to have physiologic labor. We do not want to take oxytocin or set up an IV line but the obstetricians keeps saying: "give episiotomy to prevent laceration; why did it take so long?" This makes us do physiologic labor with intervention (P 22, 41y, Doula). Different attitudes, opinions and points of view of obstetricians and midwives make them look at childbirth differently. Obstetricians see childbirth as a high-risk situation that should be taken care of under intensive care, and believe that quick interventional measures should be taken to save the mother and her baby (P14, 61y, National senior lecturer of physiologic childbirth).*

## 2. Strategies for improving the implementation of the physiologic birth program

This was another main category extracted from the interviews, and had four subcategories as follows:

**2.1. Supervising the implementation of childbirth preparation classes and physiologic childbirth.** According to the participants, one of the strategies for implementation of high-quality physiologic birth program was careful monitoring and continuation of physiologic labor implementation from the higher governmental levels.

*I feel the program isn't taken serious anymore; they used to take it serious, but not now... In the past, even the director of the hospital would come and check the shortcomings. He would check the quality of the tubs and birth balls; but now if we want to buy a birth ball, they say*

*we don't have enough budget. We didn't make any progress; we didn't even stay at the same place where we were; in fact, we regressed. Well, this program should be important to them from higher governmental levels (P2, 45y, Midwife in charge of the maternity ward).*

*Physiologic birth is disappearing. Monitoring is so important. I was invited several times and held physiologic labor workshops for midwives in the health sector; but unfortunately, there's no monitoring on to what extent they should use the things they're taught (P17, 59y, National senior lecturer of physiologic childbirth).*

*Is there any monitoring? I asked the Deputy of Treatment and I found that there isn't any. Well, what did they do? What happened to the private midwife who didn't do these things and didn't follow the instructions? Was she reprimanded? Was she notified officially to be a lesson for the others to not repeat it again? (P 1, 53y, Supervisor)*

**2.2. Supporting midwives by insurance companies.** Obstetricians believed that it is necessary to provide financial support for low-risk mothers so that they can receive physiologic labor services and have accompanying midwives. *One of the deficiencies of physiologic labor is that they don't take into account the patient. Sometimes the selection itself is inappropriate. There was a patient we knew was not a proper candidate for physiologic labor but was accompanied by a midwife until the end. But sometimes, low-risk suitable cases are deprived of these services because of financial issues (P21, 52y, Assistant professor in Obstetrics & Gynecology).*

The private accompanying midwives participating in this study believed that the government and insurance companies should support the private sector for mothers requesting private midwives: *The government should determine the insurance fees for the accompanying and private midwives; for example, they can pay 40% of the cost of these midwives. This is the greatest problem that we have in clinics. A mother asked me whether she could use her supplemental insurance, and I said that the government and insurance companies don't pay the costs of physiologic birth and private midwife (P6, 49y, Doula).*

**2.3. Holding training courses on physiologic birth.** The participants emphasized high-quality training of physiologic birth for all midwives and obstetricians. The midwife in charge of maternal health said: *We can test our instructors once a year. Then, we can hold practical retraining classes. This reminds them of their skills; we shouldn't give them a long-life certificate (P12, 54y, Maternal health officer).*

*Many more midwives should attend the training course. Most of the workshops are held privately and with very high costs. This training course should be held for all midwives who are in the public system (P20, 40y, National senior lecturer of physiologic childbirth).*

*Both gynecology residents and midwifery students should be trained for physiologic birth during their studies. There should be physiologic labor training from the residency period to change their attitude and beliefs about childbirth. We've been trained like this, but they should plan for the current students so that they can be educated based on physiologic birth, not just high-risk labor (P15, 49, Assistant Professor in Obstetrics & Gynecology).*

**2.4. Evaluation the program implementation.** In this regard, the participants believed that: *Mothers have the care booklet. In the same place where care is provided to them, an item can be added; there can be a midwife for physiologic labor training. From the pregnancy period, these trainings should be a part of their routine care (P19, 59y, National senior lecturer of physiologic childbirth).*

*Less intervention is more beneficial for the mother and the fetus; we have to decide which case needs intervention; we can separate low-risk mothers from high-risk ones and let the former undergo physiologic birth with a companion midwife (P18, 57y, Obstetrician).*

The participants emphasized increasing the number of training centers for childbirth preparation classes in the health system, creating educational space and equipment, and selecting midwives to hold classes in all health centers.

*Each midwife has to be able to train the mothers; she is in charge of them in her own health center; she shouldn't send these pregnant women to another center to receive training. Space and facilities should be provided for each center (P13, 57y, Head of the family health department). I don't do any pregnancy care for the mothers and have no information about their condition at all. That I don't know anything about the condition of these mothers is a big problem. The infrastructure needs to be modified (P16, 45y, Childbirth preparation class instructor).*

Another trainer said: *It's too early to start classes at the 20<sup>th</sup> week. Classes should be changed, both the way they're held and the beginning of them; the content also needs to be more attractive for mothers and shouldn't be too long and boring. When we say there will be 8 sessions, many mothers prefer not to come. In general, mothers like to attend a few sessions and receive the related trainings (P10, 34y, Childbirth preparation class instructor).*

According to the participants, the use of accompanying midwives was one of the strategies for the successful implementation of the program: *Given the current situation, the ratio of midwives to mothers is small. We can use accompanying midwives; it's very good for both the mother and the midwife of the maternity hospital (P5, 57y, Staff midwife).*

*Generally speaking, childbirth preparation process should not end in pregnancy; the mother who refers should be able to use this program during her pregnancy and childbirth and be accompanied by the midwife who has been with her from the beginning; or there should be several midwives so that in case one can't accompany, another can replace her (P9, 36y, Doula).*

*You can entrust all responsibilities to the accompanying midwife or the instructor of the childbirth preparation class until the time of delivery and create conditions to do the exercises with the mother; or do breathing exercises and accompany her; but specialized procedures and examinations can be performed by the midwife of the maternity hospital. In our public healthcare system, it's not yet possible for a midwife to be working in both the maternity and the health centers. Well, for this, our public system isn't so prepared yet (P11, 43y, Childbirth preparation class instructor).*

With regard to encouragement of maternity midwives to do physiologic birth, the participants said: *When birth is physiologic, midwives should spend more time and patience. They should be encouraged. As the labor is physiologic, the mother should be supported psychologically and take a warm shower. Something should be considered as allowance for midwives to increase their motivation (P7, 39y, Staff midwife). "The difficulty of a midwife's work can be considered as a motivation to perform physiologic birth; a motivation for the midwife of the maternity hospital (P4, 39y, Staff midwife).*

## Discussion

This qualitative study was conducted to explain the experiences of health care providers involved in the physiologic birth program in Iran. The experiences of these service providers

were classified into two categories of obstacles to and strategies for the implementation of physiologic birth. Five subcategories of the first category included: no continuous midwifery care in the healthcare system, lack of free accompanying midwives, lack of integrated healthcare and hospitals in service provision, low quality of childbirth preparation and implementation of physiologic birth classes, and lack of the requirements for the implementation of physiologic birth in the maternity hospital. Four subcategories of the second category included: Supervising the implementation of childbirth preparation classes and physiologic childbirth, support of midwives by insurance companies, holding training courses on physiologic birth, and evaluation of program implementation.

According to our findings, continuous midwifery care is a requirement for any healthcare system to improve midwifery services, especially the implementation of physiologic birth. Currently, there is compelling evidence from randomized controlled studies, descriptive and comparative analyses and qualitative studies regarding the benefits of continuous midwifery care for midwives, women and the health system [24–27]. According to studies conducted in Iran, this care model improves maternal and newborn outcomes [28]. Following the evidence of continuous midwifery care, the WHO's guidelines on antenatal care suggest that a midwife-led continuous care model in which a midwife or a small group of familiar midwives support a woman during the antenatal, the childbearing and postpartum periods, can be beneficial for pregnant women in environments with appropriate midwifery programs [29]. The midwifery care model is growing internationally and in countries such as New Zealand, Australia, England and Denmark [30]. Although women's access to continuous obstetric care has improved, there is no universal access to this type of care yet. Therefore, the result of this study urge that individuals at managerial and executive levels as well as policymakers and health service providers should recognize the significance of establishing such a relationship and provide the necessary resources and space to make it possible [31]. Furthermore, the WHO recommends a chosen companion during labor and childbirth [29]. The presence of accompanying midwives in the delivery wards as a simple and non-invasive intervention can lower anxiety and fear of vaginal delivery in pregnant women and improve their experiences of childbirth [32]. Various studies have been conducted all around the world to investigate the effect of accompanying midwives on the intention for having physiologic birth and improvement of maternal and newborn outcomes [33]. The participants in this study believed that in the current situation, the best strategy for implementing physiologic birth is planning to have a midwife accompanying pregnant women during the childbirth. Therefore, given the shortage in midwifery staff in maternity hospitals and the lack of continuous midwifery care in Iran, using accompanying midwives with the support of the government and insurance companies can be the best strategy for the implementation of physiologic birth.

Obviously, fear of childbirth and not bearing the pain of labor are among the most important reasons why women prefer elective cesarean section [34–36]. Given the increased rate of cesarean section around the world, international policies and approaches aim to encourage women to choose vaginal delivery [37]. Moreover, according to the evidence, participation in childbirth classes reduces childbirth anxiety and creates an appropriate response to pain [38]. Accordingly, high-quality childbirth preparation classes can significantly affect the success of implementation of the physiologic birth program [39]. Although the Iranian Ministry of Health holds childbirth preparation classes in some health centers, based on the results of the study, executive managers need to increase the number of these centers and evaluate the quality of these classes regularly. As indicated by different studies, focusing on physiologic birth methods, the deliveries supervised by a midwife can reduce unnecessary interventions in the delivery process [40] and lead to positive outcomes for both the mother and the newborn [41]. However, the midwives and obstetricians in our study considered the following as obstacles to

the high-quality implementation of physiologic birth: lack of proper physiologic birth methods, interventions by some private midwives, and the rushing midwives to do deliveries as fast as possible as they are supposed to do several deliveries in different hospitals. Therefore, strategies for the implementation of high-quality physiologic birth by private midwives may include continuous monitoring of the performance of midwives in the private sector, passing laws requiring compliance with the maximum number of deliveries, and the team cooperation of midwives in cases of labor interference. In this regard, teamwork in providing health and maternity care services will lead to high-quality services [42].

The results of our study indicate that when it comes to maternal health, one of the challenges is the different views of midwives and obstetricians regarding the physiologic or pathologic process of pregnancy and childbirth, causing contradictions in providing services to pregnant mothers. The educational approach in which gynecology residents and specialists view pregnancy and labor as high-risk processes calls for intervention in the physiological process of childbirth to make it faster since it is a high risk situation. One of the challenges of physiologic birth in Iran, according to Jenani et al., was the absence or low quality of physiologic birth education for midwifery and medical students, residents, and obstetricians [43]. A systematic review of the barriers to and facilitators of the physiological approach to labor and delivery shows that rather than evidence-based guidelines that recommend a physiological approach, a risk-based approach is taught and implemented in obstetrics and gynecology departments, which may act as an obstacle to the physiological process of childbirth [44].

A survey on the role of midwives in the professional delivery team revealed that they are independently responsible for the care provided during vaginal delivery and use their knowledge and experience along with the expertise of the midwifery team in this regard. However, the responsibility and autonomy of midwives were somewhat undermined by obstetricians and obstetricians, even when they provided care during vaginal delivery [45]. Obstetricians and obstetricians should trust the evidence-based knowledge and competence of midwives in doing vaginal delivery and refrain from intervention [46]. Therefore, based on the results of this study, in order to change their attitude, education should be directed towards the physiological methods of childbirth under the guidance of a midwife from the residency period, and only in case of high-risk maternal and fetal conditions, should the obstetricians be called and the necessary intervention be performed.

Discontinuous physiologic birth services (childbirth preparation classes and processes) decrease mothers' unwillingness to attend these classes and lower the motivation of midwives and instructors for holding childbirth preparation classes [47]. According to studies, one of the reasons that mothers attend prenatal education classes is to improve the outcome of childbirth and have a positive childbirth experience [48, 49]. Thus, if these classes do not improve the outcomes of childbirth, they are not appropriately welcomed by the mothers. Additionally, the midwives and the instructors of physiologic birth classes who participated in this study also confirm that they will have more motivation to implement quality classes and communicate with the mother regularly provided that they can play an effective role in improving the outcomes of delivery by performing physiologic birth methods [50, 51]. The results of this study showed that the current status of the physiologic birth program in Iran where classes are held only in some health centers and the referring mothers to those centers and lack of birth plan is one of the biggest obstacles to the promotion of physiologic birth. Therefore, based on our study, there is a need to plan for the continuous implementation of the physiological delivery program in the pregnancy and delivery process. Due to the fact that at the time of conducting this study, the COVID-19 pandemic in Iran was almost controlled by nationwide vaccination and that in person physiologic childbirth classes were being implemented, this study did not have any limitations in this regard.

## Conclusion

According to the experiences of the health providers involved in the physiologic birth program, health planners and policymakers should provide the ground for the implementation of this type of labor by removing the obstacles and providing operational strategies needed in Iran. According to the results of our study, a number of measures can be taken in order to promote implementation of this program. These included setting the stage for physiologic birth in the system that is responsible for the supply of healthcare providers (supply of manpower, equipment, facilities, and program implementation), building separate low-risk and high-risk wards in maternity hospitals, professional autonomy of midwifery services, training childbirth health providers (midwives and obstetricians) on physiologic birth, monitoring the quality of implementing the physiologic birth program, and insurance support of midwifery services.

## Supporting information

**S1 Checklist. Consolidated criteria for reporting qualitative studies (COREQ): 32-item checklist.**
(DOCX)

## Acknowledgments

Thanks to all the participants who shared their experiences with us.

## Author Contributions

**Conceptualization:** Azam Moridi, Parvin Abedi, Mina Iravani, Shala Khosravi, Narges Alianmoghaddam, Elham Maraghi, Najmieh Saadati.

**Data curation:** Azam Moridi.

**Formal analysis:** Azam Moridi, Parvin Abedi, Narges Alianmoghaddam, Elham Maraghi.

**Funding acquisition:** Parvin Abedi.

**Investigation:** Mina Iravani.

**Methodology:** Azam Moridi, Parvin Abedi, Mina Iravani, Shala Khosravi, Narges Alianmoghaddam, Najmieh Saadati.

**Project administration:** Azam Moridi, Shala Khosravi.

**Software:** Azam Moridi, Elham Maraghi.

**Supervision:** Parvin Abedi, Mina Iravani, Shala Khosravi, Narges Alianmoghaddam, Elham Maraghi, Najmieh Saadati.

**Validation:** Azam Moridi, Najmieh Saadati.

**Writing – original draft:** Azam Moridi, Parvin Abedi.

**Writing – review & editing:** Azam Moridi, Parvin Abedi, Mina Iravani, Shala Khosravi, Narges Alianmoghaddam, Elham Maraghi, Najmieh Saadati.

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
