## [Decision Letter · Decision Letter 0]

13 Dec 2022

PONE-D-22-26862Experiences of health providers regarding implementation

of the physiologic birth program in Iran: A qualitative

content analysisPLOS ONE

Dear Dr. Abedi,

Thank you for submitting your manuscript to PLOS ONE. After careful consideration, we feel that it has merit but does not fully meet PLOS ONE’s publication criteria as it currently stands. Therefore, we invite you to submit a revised version of the manuscript that addresses the points raised during the review process.

Specifically:A description of the coding tree and providing theme(s)

We look forward to receiving your revised manuscript.

Kind regards,

Forough Mortazavi

Academic Editor

PLOS ONE

Journal Requirements:

“The expenses of this study were provided by Ahvaz Jundishapur University of Medical Sciences. The funder did not play any role in design, data collection, data interpretation, writing and submitting the manuscript to a journal.”

Additional Editor Comments:

Dear authors,

Thank you for submitting the manuscript to PLOS ONE. This manuscript is dealing with an important subject. Therefore, to improve it and receive deserved attention from the readers, some revisions are needed.

Introduction

Measures stated in the first three lines of page 6 are related to making childbirth a satisfactory process and are not necessary physiologic birth. So, with this distinction in mind, PLS provide a precise definition of the physiologic birth.

PLS clarify if the midwives working in a hospital are allowed to serve simultaneously as a private midwife and as personnel on duty on the same shift.

PLS describe the impact of the COVID-19 pandemic on the holding of childbirth preparation classes in the country at the time of the study.

Methods:

PLS describe the rationale for the recruitment method. (Interview vs. focus group discussion)

How many people refused to participate or dropped out? Reasons?

How many data coders coded the data?

Results

PLS provide a description of the coding tree.

Discussion

PLS discuss the generalizability of the results.

PLS explain the probable effects of the COVID-19 pandemic on your results.

PLS check the text especially the results section and quotes for grammatical errors

PLS use the COREQ checklist, or other relevant checklists listed by the Equator Network, to ensure complete reporting of the study and attach the completed checklist after revising the manuscript.

PLS discuss limitations and potential sources of bias.

Reviewers' comments:

Reviewer's Responses to Questions

**Comments to the Author**

1. Is the manuscript technically sound, and do the data support the conclusions?

Reviewer #1: Yes

Reviewer #2: Yes

Reviewer #3: Yes

2. Has the statistical analysis been performed appropriately and rigorously? 

Reviewer #1: Yes

Reviewer #2: Yes

Reviewer #3: Yes

3. Have the authors made all data underlying the findings in their manuscript fully available?

Reviewer #1: Yes

Reviewer #2: Yes

Reviewer #3: Yes

4. Is the manuscript presented in an intelligible fashion and written in standard English?

Reviewer #1: Yes

Reviewer #2: Yes

Reviewer #3: Yes

5. Review Comments to the Author

Reviewer #1: This is a valuable paper, but need some revisions as below:

1-It is better to mention the subcategories in the abstract

2-In the obtaining consent form section, write both the consent to participate in the study and the consent to record the interviews were obtained.

3-Please mention what was your strategy in case you were not allowed to record the interviews?

4-In purposive sampling, you should mention the purposive sampling with maximum variation in term of some variable such as duration of experience, .....

5-please mention the fully characteristics of the people participating in the research after each meaning unit. ex( p10, 32y, midwife)

6-You can merge the categories in one theme

7- the subcategory entitled: Supervision, monitoring and continuous evaluation of the quality of childbirth preparation and physiologic birth classes is too long, please change it

8-in subcategory” Assessing program implementation”, Can't you use the word evaluation instead assessing???please explain your reasons

9- participants did not mention the effect of COVID-19 on the quality of participation of mothers in classes?

Reviewer #2: Dear Authors

Thank you very much for writing a valuable article, and I have just one minor revision comment on the manuscript:

-In abstract:

The year and place of the study should be stated

I accept this manuscript.

Best regards

Reviewer #3: Thank you for the opportunity to review this valuable manuscript.

The prevalence of cesarean delivery is much higher compared to vaginal delivery, In Iran. One of the most basic reasons is the cultural issue and the attitude of the health providers.

This research has valuable results to improve the health of the Iranian society and considering the important role of the type of delivery in the health of the mother, infant and society, I recommend publishing the manuscript. The results of this manuscript are very valuable and needed by the Iranian society.

With respect

6. PLOS authors have the option to publish the peer review history of their article (what does this mean?). If published, this will include your full peer review and any attached files.

Reviewer #1: No

Reviewer #2: No

Reviewer #3: No

---

## [Author Response · Author response to Decision Letter 0]

4 Feb 2023

Response to reviewer’s comments

Dear Editor

Re: Manuscript 

Greetings and Regards

Please find attached a revised version of our manuscript “[Experiences of health providers regarding implementation of the physiologic birth program in Iran: A qualitative content analysis]”, after careful assessment of editor and reviewers' comments [PLOS ONE]. 

Your comments and those of the reviewers were highly insightful and enabled us to greatly improve the quality of our manuscript. In the following pages please see our point-by-point responses to each of the comments of editor and the reviewers.

We hope that the revisions in the manuscript and our accompanying responses will be sufficient to make our manuscript suitable for publication in [PLOS ONE].

Yours sincerely,

Prof Parvin Abedi

Midwifery Department

Ahvaz Jundishapur University of Medical Sciences

 

Thank you. All requirements of journal was checked and corrected in the paper. 

These statements were added to the manuscript and cover letter. 

Thank you. Added. 

“The expenses of this study were provided by Ahvaz Jundishapur University of Medical Sciences. The funder did not play any role in design, data collection, data interpretation, writing and submitting the manuscript to a journal.”

Thank you. Our statement about funding is same as above and we included it in the cover letter. 

Thank you. The following statement was added to the manuscript: 

In order to preserve participants' confidentiality, and according to the requirements of the METC Groningen in which anonymity of participants must be guaranteed, we are not willing to share the qualitative datasets (the interview transcripts) in the main paper or additional supporting files. We cannot share the data due to ethical restrictions that the data contains potentially identifiable and sensitive information of the participants. Although we did remove personal identifiers from the interview transcripts (e.g. names and addresses), the transcripts are likely to contain references to the contextual identifiers in individual stories and make individuals identifiable. When providing their informed consent to participate in the study, participants were ensured their privacy would be protected. They did not provide consent for their data to be shared in a repository. We can provide access to the transcripts and audit trail on request and subject to certain conditions. Data requests must be addressed to the Reproductive Health Promotion Research Center of Ahvaz Jundishapur University of Medical Sciences, that will provide access after evaluating requests: RHPRC@ajums.ac.ir.

Thank you. The references were reviewed and none of them were retracted or had a correction. 

Additional Editor Comments to the Author:

Dear authors,

Thank you for submitting the manuscript to PLOS ONE. This manuscript is dealing with an important subject. Therefore, to improve it and receive deserved attention from the readers, some revisions are needed.

Response: Thank you 

Introduction

Measures stated in the first three lines of page 6 are related to making childbirth a satisfactory process and are not necessary physiologic birth. So, with this distinction in mind, PLS provide a precise definition of the physiologic birth.

Response: [it was added and highlighted, page 6]

PLS clarify if the midwives working in a hospital are allowed to serve simultaneously as a private midwife and as personnel on duty on the same shift.

Response: [it was explained and highlighted it]

PLS describe the impact of the COVID-19 pandemic on the holding of childbirth preparation classes in the country at the time of the study.

Response: [The impact of the COVID-19 epidemic on childbirth preparation classes was mentioned in the introduction. Due to the fact that at the time of conducting this study, the COVID 19 pandemic in Iran was almost controlled by nationwide vaccination and that face-to-face physiologic childbirth classes were being implemented, this study did not have any limitations in this regard.]

Methods

PLS describe the rationale for the recruitment method. (Interview vs. focus group discussion)

Response: [It was added.

Due to the maximum variety of the participants who were from different levels of service providers, including managerial, executive, clinical, and educational levels in both public and private sectors, the interviews were expected to provide a deep understanding of the phenomenon of physiological childbirth from the participants’ point of view.]

How many people refused to participate or dropped out? Reasons?

Response: [No participant was excluded from the study.]

How many data coders coded the data?

Response: [The coding process was performed by two authors (AM, PA).]

 Results

 PLS provide a description of the coding tree.

Response: [Added]

 Discussion

 PLS discuss the generalizability of the results.

Response: [Added]

PLS explain the probable effects of the COVID-19 pandemic on your results.

Response: [Added]

 PLS check the text especially the results section and quotes for grammatical errors

Response: [The manuscript was edited by a person who is expert in English literature]

 PLS use the COREQ checklist, or other relevant checklists listed by the Equator Network, to ensure complete reporting of the study and attach the completed checklist after revising the manuscript.

Response: [Done.]

 PLS discuss limitations and potential sources of bias.

Response: [There was no limitation in the implementation of this study.]

Reviewer Comments to Author

Reviewer: 1

 1-It is better to mention the subcategories in the abstract

Response: [Added]

 2-In the obtaining consent form section, write both the consent to participate in the study and the consent to record the interviews were obtained.

Response: [Thank you. Added.] 

 3-Please mention what was your strategy in case you were not allowed to record the interviews?

Response: [The participants’ consent for recording the interview was obtained, and in case they did not allow the recording, field notes were taken.]

 4-In purposive sampling, you should mention the purposive sampling with maximum variation in term of some variable such as duration of experience, .....

Response: [Purposive sampling was done considering maximum variety in terms of the participants’ workplace, work experience (years), educational attainment, and age.] 

5-Please mention the fully characteristics of the people participating in the research after each meaning unit. Ex (p10, 32y, midwife)

Response: [Done]

6-You can merge the categories in one theme

Response: [Considering the importance and breadth of the category, we preferred to keep the division in two themes.]

7- The subcategory entitled: Supervision, monitoring and continuous evaluation of the quality of childbirth preparation and physiologic birth classes is too long, please change it

Response: [Done.]

8-in subcategory” Assessing program implementation”, Can't you use the word evaluation instead assessing???please explain your reasons

Thank you. The word "assessing" was replaced by evaluation. 

9- participants did not mention the effect of COVID-19 on the quality of participation of mothers in classes?

Response: [ The participants mentioned the problems of the online classes of physiological childbirth, which were categorized in the category of low quality of the classes.]

Reviewer: 2

 In abstract: The year and place of the study should be stated

Response: [Added.]

---

## [Decision Letter · Decision Letter 1]

1 Mar 2023

Experiences of health providers regarding implementation

of the physiologic birth program in Iran: A qualitative

content analysis

PONE-D-22-26862R1

Dear Dr. Abedi,

We’re pleased to inform you that your manuscript has been judged scientifically suitable for publication and will be formally accepted for publication once it meets all outstanding technical requirements.

Kind regards,

Forough Mortazavi

Academic Editor

PLOS ONE

Additional Editor Comments (optional):

Reviewers' comments:

Reviewer's Responses to Questions

**Comments to the Author**

1. If the authors have adequately addressed your comments raised in a previous round of review and you feel that this manuscript is now acceptable for publication, you may indicate that here to bypass the “Comments to the Author” section, enter your conflict of interest statement in the “Confidential to Editor” section, and submit your "Accept" recommendation.

Reviewer #1: All comments have been addressed

Reviewer #2: All comments have been addressed

2. Is the manuscript technically sound, and do the data support the conclusions?

Reviewer #1: Yes

Reviewer #2: Yes

3. Has the statistical analysis been performed appropriately and rigorously? 

Reviewer #1: Yes

Reviewer #2: Yes

4. Have the authors made all data underlying the findings in their manuscript fully available?

Reviewer #1: Yes

Reviewer #2: Yes

5. Is the manuscript presented in an intelligible fashion and written in standard English?

Reviewer #1: Yes

Reviewer #2: Yes

6. Review Comments to the Author

Reviewer #1: (No Response)

Reviewer #2: Dear Authors

Thank you very much for writing a valuable article.

I accept this manuscript.

Best regards

7. PLOS authors have the option to publish the peer review history of their article (what does this mean?). If published, this will include your full peer review and any attached files.

Reviewer #1: No

Reviewer #2: No

---

## [Editor Report · Acceptance letter]

6 Mar 2023

PONE-D-22-26862R1 

Experiences of health providers regarding implementation of the physiologic birth program in Iran: A qualitative content analysis 

Dear Dr. Abedi:

I'm pleased to inform you that your manuscript has been deemed suitable for publication in PLOS ONE. Congratulations! Your manuscript is now with our production department. 

Kind regards, 

on behalf of

Dr. Forough Mortazavi 

Academic Editor

PLOS ONE